# Can Smart Contracts Become Smart?
# An Overview of Transaction Impact on Ethereum DApp Engineering

Emanuel Onica
emanuel.onica@uaic.ro
Alexandru Ioan Cuza University
Iaşi, Romania

Marius Georgică
georgica.marius756@gmail.com
Alexandru Ioan Cuza University
Iaşi, Romania

## ABSTRACT

Despite the growth in the number of decentralized applications (DApps) supported by the Ethereum blockchain, we can observe the narrow scope of these DApps, concentrated within the fintech and games areas. A cause for the lack of range of DApps lies in the fees for transactions sent to backing smart contracts. While consistent steps have been made to overcome cost efficiency problems, introducing rollups as a secondary layer solution, intertwined accessibility and security drawbacks still persist. Measures addressing some of these issues like account abstraction were independently proposed. These solutions bring changes in transaction handling that often exceed the scope of smart contracts, where the core of DApp logic resides. Integrating such measures often requires the use of new frameworks and understanding the changes in the transaction flow, which can prove challenging to a DApp developer. A question is whether the current landscape of solutions proposed for increasing usability is capable of producing a consistent impact on DApp scope trends. In this position paper we try to answer this, raising also the matter of impact on DApp engineering.

## CCS CONCEPTS

• **Software and its engineering** → **Software design engineering**; • **Computer systems organization** → *Distributed architectures.*

## KEYWORDS

DApps, Smart Contracts, Ethereum

**ACM Reference Format:**
Emanuel Onica and Marius Georgică. 2023. Can Smart Contracts Become Smart? An Overview of Transaction Impact on Ethereum DApp Engineering . In *4th International Workshop on Distributed Infrastructure for the Common Good (DICG '23), December 11–15, 2023, Bologna, Italy*. ACM, New York, NY, USA, 6 pages. https://doi.org/10.1145/3631310.3633492

## 1 INTRODUCTION

Like most blockchain networks, Ethereum's purpose is to maintain a ledger formed of securely chained blocks. The essential ensured properties are decentralized trust on the validity of transactions updating this ledger, as well as its transparency and immutability. Ethereum was the first blockchain to support Turing complete smart contracts, small programs executed by the Ethereum Virtual Machine (EVM) on the network nodes. Transactions are initiated by Externally Owned Accounts (EOAs) associated with the users, and can either update the balance of other EOAs or the state of smart contracts. The most used high-level language for programming Ethereum smart contracts is Solidity, which is compiled into a low-level bytecode executed by the EVM. A Solidity function call that modifies the state of an EOA or contract essentially corresponds to an Ethereum transaction.

Are smart contracts smart enough? This is a question that pops back periodically in the online media either on topics related to the utility of smart contracts or in the context of security issues [2, 3, 12]. The initially foreseen utility of smart contracts introduced by Ethereum was extremely versatile, covering a wide range of use cases: currency exchanges, financial derivatives, insurance claims and settlements, games, supply chain tracking, land registries, and many others [1]. All these are supposed to benefit from the decentralized trust guaranteed for the executed transactions. The introduced decentralized application model, referenced as *DApp*, typically consists of an augmented web application where the frontend functionality is capable of transaction initiation and the backend provides the support of calling smart contracts. Currently, the bulk of Ethereum transactions is generated by DApps whose scope is narrowed to fintech and games, including gambling. Many of these DApps use *tokens*, which are essentially DApp specific currencies (fungible) or unique assets (non-fungible) that can be traded, exchanged, and used in internal DApp operations. The concentration of range in the mentioned areas was confirmed by periodical overviews of the Ethereum DApps landscape, for more than half of counted DApps [5, 31]. This makes the question at the beginning of this paragraph seem rhetorical. Clearly, Ethereum smart contracts have not proven smart enough yet to gain traction in all potential areas of application. The more important question is why?

We believe the answer is less related to the expressiveness of smart contract logic and more to the fees charged for transactions initiated toward smart contracts. This has a direct negative impact on the usability of DApps. Measures for increasing transaction cost efficiency and DApp accessibility were proposed, which partially eliminate the fee burden. Currently, some of the most prominent

solutions are rollups and account abstraction. To integrate such solutions a DApp developer must resort to intricate patterns touching the transaction flow, which often exceed the scope of smart contracts that hold the core of DApp logic. We believe that smart contracts cannot become smart enough to produce an impact on DApp trends unless accompanied by such changes in the typical transaction flow. This can bring, however, a consistent added cost to the DApp engineering complexity.

In this paper, we advocate our position. Section 2 provides background on the costs of transactions to Ethereum smart contracts. In Section 3 we overview measures introduced to circumvent efficiency and accessibility issues in DApps in direct relation to lowering transaction fees, and their impact on the DApp engineering. In particular, we introduce a simple use case example of an e-gov DApp outside the fintech and games realms, to show the viability of rollups. In Section 4 we discuss additional details on why the mechanisms we overview do not seem yet to have a significant impact on DApp diversity. Finally, we conclude in Section 5, where we offer a perspective on possible future improvements.

## 2 TRANSACTION COSTS BACKGROUND

Each transaction to an Ethereum smart contract or for creating a contract has a cost. This prevents network abuse by limiting potentially malicious invocation of smart contract code, which could lead to DoS attempts. The computational transaction cost is defined in the Ethereum specifications and is quantified in units of *gas*. In essence, the total cost of a transaction can be split in two parts: the cost of submitting the transaction and the cost of executing the transaction.

The cost of submitting a transaction starts from a fixed gas amount, which depends on the transaction purpose: either a regular transaction or one used to instantiate a new contract. This amount is incremented per each byte encoded in the transaction call, with a higher cost for non-zero bytes. The cost of executing the transaction sums up the gas cost of each low-level instruction executed by the EVM when running the corresponding smart contract function. The Ethereum specifications periodically changed the defined execution costs, but storage typically inflicts substantially higher amounts.

A transaction initiator must pay a fee expressed in Ether (ETH), the cryptocurrency of Ethereum. The fee is computed as a product of the transaction cost multiplied by the price per unit of gas. This price is composed of a base fee and a priority fee as an incentive for transaction validators. The price per unit of gas varies, depending on the network load.

## 3 TRANSACTIONS AND DAPPS USABILITY

Various solutions have been proposed in the Ethereum landscape for increasing usability. One direction is to attempt scaling the transaction processing load, which has a direct impact on lowering transaction fees. Another direction focuses on accessibility, by making it possible for users to pay transactions with other currencies, or even exempt them from payment altogether. In the following, we overview two prominent solutions in each category, namely rollups and respectively account abstraction. We observe how complicated is their integration with DApps and provide a use case example for what we find as the most facile approach.

### 3.1 DApps and rollup transactions

Solutions grouped under the Layer 2 (L2) umbrella, consist of horizontal scaling by extending the processing power of the main Ethereum network (L1) with secondary networks capable of transaction execution. L2 networks come in flavors that differ in how they interact with L1 for deriving trust guarantees.[1] Currently, the most popular L2 solutions are rollups, which are the only ones to store transaction data on the main Ethereum network, providing a higher degree of trust. Essentially, besides executing transactions, nodes in rollup networks batch multiple transaction data into single "rollup" transactions submitted to L1.

Rollups come in two variants: *optimistic rollups* and *zk-rollups*. Optimistic rollups consider transactions valid implicitly, posting on L1 their compressed data. Transactions can be challenged during a certain time window using fraud proofs. Networks that implement the optimistic rollup pattern typically run EVM compatible nodes. This means that the backend contracts of a typical Ethereum DApp can be ported to the rollup network with minimal or no changes.

Zk-rollups use zero-knowledge proofs for validating transactions, offering stronger security guarantees. The transaction summary is posted on L1 along with the cryptographic information required for verification. However, unlike optimistic rollups, most zk-rollups are not EVM compatible. Computing zero-knowledge proofs for a transaction to a generic smart contract that can use any native EVM instruction is computationally difficult. Early solutions for zk-rollups (Loopring [19], zkSync Lite [24], dYdX [14]) restrict the use of smart contracts to specific subsets of transactions, in particular for transactions with tokens. Other platforms like Starknet [22] introduced new running environments and specific programming languages (Cairo [13]), providing support for expressive zk-provable smart contracts. The above either limits severely the use cases for integrating with an Ethereum DApp or might require breaking changes when porting existing smart contracts. One approach for more complex DApps can be splitting the contracts set depending on the transaction support. Token contracts could be deployed on the zk-rollup, while code corresponding to other unsupported transactions could be grouped in contracts deployed on L1. The DApp frontend would selectively route transactions. This approach, however, would only partially reduce DApp usage fees.

Recent zk-rollup solutions make use of *zkEVMs* runtime implementations that aim to be compatible with EVM, which should allow running any contract. Still, these solutions are either in alpha or beta stages (Linea [18], Polygon-zkEVM [21]) or require different Solidity compilers (zkSync Era [23]).

Both optimistic and zk-rollups present a common drawback. A centralized *sequencer* is typically the component tasked with ordering transactions, building blocks, and submitting these to L1. The sequencer is under the control of the rollup operator and is usually the sole manager of an L2 transaction mempool. Even though this does not have a direct impact on DApp development, it brings concerns about potential censorship or unfair handling of transactions.

---

[1]We do not consider as part of L2, solutions like sidechains, which only provide bridges to Ethereum and their security is completely independent.

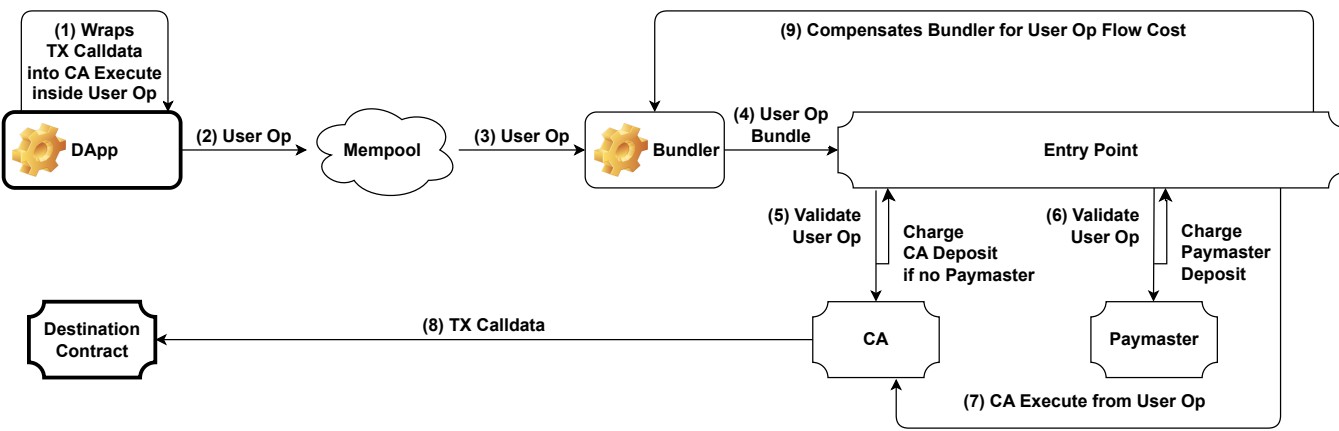

**Figure 1: High level overview of the User Operation flow in ERC-4337**

## 3.2 DApps and account abstraction

The need for transaction payment complicates the onboarding of new users. A DApp that initiates transactions on behalf of the user would normally require the user to create an Ethereum externally owned account (EOA), fund this account with Ether, and then sign and pay for the transaction. Creating an EOA and signing the transaction are steps that can be handled transparently by a DApp frontend. Funding the account with Ether is not. For this, a user would typically need to register with a cryptocurrency exchange, get through a KYC process, and acquire the needed Ether. This already implies starting a learning curve for new users who do not have any knowledge of Ethereum economics.

Then, the user might not be comfortable transferring funds between multiple accounts if using multiple DApps, because transactions have a cost even between the user's accounts. Therefore, typically users end up installing a wallet like MetaMask or Coinbase as a typical way to manage an EOA and its associated funds for different payments. DApp frontends should then integrate with external wallet implementations, for obtaining the required signature when initiating the transaction. This creates some dependency between the DApp frontend and the wallet implementation. However, wallets typically adhere to a standardized RPC API for Ethereum interaction [27], supposed to minimize conflicting behavior when switching between different implementations.

The major problem remains the need for Ether for transaction payment, which most users are not familiar with. Moreover, for some use cases like e-gov land registries or voting DApps that address large categories of people and infrequent usage, it is hard to expect that users would be willing to pay for transactions. A pattern that is a possible solution for this problem is meta-transactions. These are mechanisms that decouple the user transactions from signing and paying with an EOA's credentials. The transaction calldata is sent as a user signed message to a relayer, which further envelopes it in a meta-transaction that preserves user identity but can be countersigned and paid by a different entity. Multiple variations of this pattern were developed in recent years [7, 10, 11, 25], but ERC-4337: *Account Abstraction* [4] emerged as a proposal for Ethereum standardization.

ERC-4337 introduces a message format that wraps the transaction calldata, named *User Operation*. We summarize its path in Figure 1. The DApp frontend is supposed to send the User Operations to a canonical mempool, separate from the normal Ethereum transaction mempool. From there, a new entity type: *bundlers*, select and pack multiple User Operations in bundles and pass these via a transaction to a standard defined EntryPoint contract deployed on the Ethereum network. This triggers a flow of interactions with the final purpose of executing the original transaction's calldata. The EntryPoint first validates each User Operation with a corresponding *contract account* (CA). The CA is a smart contract encoded as a sender in a User Operation that basically replaces the EOA. The main reason for introducing this CA in the flow is to permit the implementation of customized verification logic of the User Operation, e.g., multiple signatures or other patterns, instead of verifying the standard EOA signature. Further, the EntryPoint flow permits an optional paymaster contract to validate the User Operation and to pay for it. The EntryPoint charges either the CA or the paymaster for the User Operation and calls an execution method in the CA for the User Operation calldata. This execution method in the CA further calls the final intended destination smart contract function. Finally, the EntryPoint compensates the bundler, which paid for the transaction that initiated this entire flow.

The proposed ERC-4337 transaction flow has the major advantage of simplifying end-users interaction, exempting them from paying for transactions when these can be sponsored by a third party. However, the whole process is actually more complex than the above simplified description. We identify several aspects that can have an impact on DApp engineering.

The calldata originally wrapped in the User Operation by the DApp frontend represents the transaction to a generic execution function in the deployed CA, further calling the intended destination smart contract. From the DApp frontend perspective, this calldata is not directly the call to the function in the destination smart contract as in a regular transaction case. ERC-4337 defines the interface for the User Operation validation function that should be implemented by the CA but does not specify an interface for an execution function. In order to support building and sending User

Operations compatible with an ERC-4337 CA instead of direct transactions to backend smart contracts, the DApp frontend must be aware of the execution function signature in the CA. Otherwise, a user whose CA does not respect the execution signature supported by the DApp frontend, will not be able to use the DApp. One might argue that this is similar to ensuring different compatibility with different normal wallets, where the DApp frontend should adapt in order to support various implementations. However, normal wallets do not need to interfere with building a transaction to a smart contract, which is typically performed using a dedicated library. The coupling is tighter when using a CA, which can introduce different calldata formats, depending on the execution function the user's CA implements. A solution for this issue is present in ERC-6900 [6], meant to further standardize the possible composable logic in CA implementations. ERC-6900 is a relatively complex specification touching multiple patterns built around ERC-4337, but essential to our discussion is that it defines a standard interface for the execution methods of a CA. The issue that remains is that not all CA implementations adhere to this specification yet. ERC-6900 is more recent than ERC-4337, and a diverging code base used for deployed CAs already appeared before its release. Handling this variation can be a complicated challenge in a DApp frontend implementation.

An issue that is somewhat similar to the case of the rollups transaction flow via centralized sequencers concerns the User Operations mempool. ERC-4337 states that User Operations should be directed to a public mempool, avoiding trust assumptions on the bundlers selecting these User Operations. At the date of writing of this paper this public canonical mempool is still under development, although foreseen to be operational in the near future [15]. Currently, the ERC-4337 ecosystem operates using private mempools, with User Operations being sent directly to bundler nodes. This assumes the DApp trusts the bundler for not censoring User Operations.

For the moment, a DApp frontend could periodically switch between different bundlers when submitting User Operations. ERC-4337 defines a standard RPC API that is used for interaction with bundler nodes. However, various bundler implementations define extra methods. Therefore, a DApp that changes bundlers for sending User Operations has to take into account such possible differences.

At the time of writing both ERC-4337 as well as ERC-6900 are still in draft status. The networking specification, which should contribute to the operation of the canonical mempool is still a work-in-progress. However, the proposals do not change the native protocol run for normal transactions by the Ethereum network nodes. Therefore, multiple implementations for the components emerged and applications already started adopting the account abstraction pattern. There are various aspects as the ones we mentioned above that we find yet volatile for expecting stable DApp implementations. Nevertheless, we believe this to be a very promising direction towards an improvement in the DApp range.

## 3.3 Use case evaluation

Following our overview, optimistic rollups currently seem the most accessible solution for a developer faced with choosing a design that improves the chances of adoption for a DApp. We implemented a simple use case scenario for testing our assumption, and also to verify the efficiency improvement. We chose Optimism [20], which

is a rollup currently ranked among the highest in terms of total currency value locked.

We opted for an e-gov DApp used for managing land registries, which is outside the dominating range of fintech and games. Despite the apparent lack of popularity, e-gov DApps represent an important area where decentralized trust and immutability have useful applications. A statistic lists 203 e-gov blockchain related initiatives in over 40 countries, out of which 19 are concerned with land registry management [8].

We used a single smart contract in our example implementation, where functions are focused on storing information, the most costly part in terms of gas. The users of the DApp have two possible roles: government staff charged with adding new data about land and people who own the land, can transfer it by changing ownership and lease it. The smart contract has a simple data flow, implemented mainly through seven functions. Two functions are used for registering and respectively removing government staff credentials, retained in a map as EOA addresses of 20 bytes in length. One function, restricted only to government staff, is used for registering new land data like ownership title information, owner information and geographical position. This uses six dynamically-sized strings, one EOA address and a boolean flag. Two functions are used for initiating changing land ownership by government staff, and respectively for responding to the change by owners. The first stores temporary information about the pending ownership change, while the second frees that information and settles the change. Two more functions have a similar role but for leasing land properties.[2]

The frontend of the DApp was implemented using the Angular framework. We used the *ethers.js* [16] library for sending transactions and querying the smart contract. The frontend integrated MetaMask [17] as an EOA wallet, used to sign and pay the transactions. We deployed our backend smart contract on both Goerli test networks that mimic the Ethereum main network and respectively the Optimism rollup network. The frontend of the DApp, in particular the transaction initiation, did not require any implementation changes when porting the application between the two networks. This is what we expected since Optimism is EVM compatible.

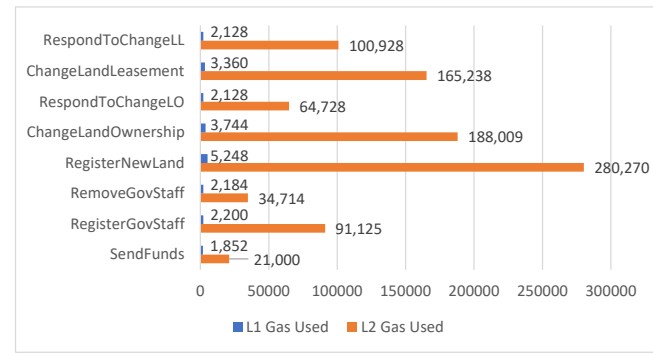

**Figure 2: Gas cost split on L1 and L2 for a transaction executed on L2**

---

[2]We note that our implementation had also a didactical purpose of demonstrating a DApp usage. Therefore, we are aware of some redundancies in its functionalities, which might seem irrelevant from an efficiency perspective.

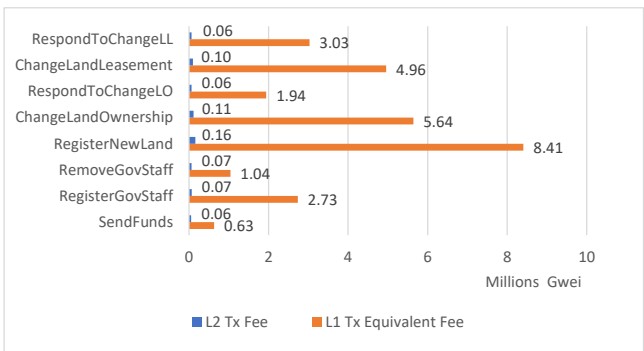

**Figure 3: Transaction fees on separate L1 and L2 executions**

The transaction fee on Optimism is computed similarly to the main Ethereum network, as the product of the transaction cost in gas with the price per gas unit. However, to this L2 fee, an L1 cost is added for the portion of the transaction data submitted as the rollup to the main network. The cost benefit for a user results from the much lower price per gas unit on L2 compared to L1. We evaluated this for our use case application example.

First, we measured the amounts of gas used by each contract method on each of the layers, using the Optimism SDK for L2. In addition, we also measured the gas used by a simple currency transfer, which corresponds to the minimal cost of a transaction. The results are depicted in Figure 2. The transaction for new land registration is the most costly since it has the highest storage requirement. Most importantly, we notice that the more expensive part of the gas cost on L1 is on average only 2.5% of the total.

At the time of our evaluation, the price per gas unit on L1 oscillated around 30 Gwei, and on L2 around 0.002 Gwei.[3] In Figure 3 we compared the fees when executing on L2 with the fees when executing the transaction completely on L1.

At the time of our experiments, the fiat exchange rate of Ethereum varied around 1800$ per Ether, which means 1.8$ per 1M Gwei. Most transactions in our e-gov use case would be related to registration and changes of land properties, which would average to approximately 4.8M Gwei per transaction in the L1 case, equivalent to a price of 8.64$. If executed on L2, the average is approximately 0.1M Gwei per transaction, which results in a price of 0.18$. With the caveat that both the exchange rate and the gas price are volatile metrics, the savings at the evaluated rates are 48 times on average for the most frequent transactions. We are aware that our experiment is a simplistic one, but in an effective case of an e-gov DApp built for production, we expect that smart contract functions would benefit from further optimization for their storage needs.

## 4 DISCUSSION

From our overview, the mechanisms implemented in the transaction flow might seem to alleviate the transaction fee issue and favor DApp usability. However, we cannot see yet a significant increase in the DApps diversity. One reason is that presented mechanisms

---

[3]1 Ether = $10^9$ Gwei. The values considered correspond to the beginning of April 2023, when we started our experiments. We note that the gas price on L2 increased after the Optimism Bedrock update in June 2023, but was still orders of magnitude lower than on L1.

are recent and their full impact is yet to be observed. We discuss several other motives that might still limit the DApps range.

Another main cause is precisely the changes that appear in the transaction flow and the difficulty of integration in DApps for use cases not previously tested. This is particularly the case when using the transaction model defined in ERC-4337, which comes along with multiple new logic constructs on the smart contracts side, like integrating the CA or the paymaster. For rollups, in general situations like the example provided in the previous section, the transaction changes are transparent and the DApp integration is seamless, without mandatory changes in the smart contract logic. However, this is not the case when the DApp requires explicit communication or sending funds between L1 and L2, which requires different specific flows. Moreover, rollups and the ERC-4337 account abstraction might seem to a developer to be divergent technologies. Both include specific nodes in their architecture, i.e., sequencers and respectively bundlers for which interoperation might seem questionable. Indeed, the ERC-4337 authors raised the issue of the need for a new RPC request that sequencers should support in order to conditionally accept transactions from bundlers [28]. Otherwise, these transactions are prone to revert due to delays from validation in the User Operation path. Arbitrum, the most used rollup network, already offers support for this change via a new RPC endpoint [30]. It is unclear yet if Optimism offers a similar solution but this is brought into discussion [26].

Privacy and lack of trust in transactions have been concerns since the early days of Ethereum. Because Ethereum is a public blockchain transactions are inherently available in the public domain, and ensuring their privacy is a complicated matter. Discussing solutions is out of the scope of this paper but we acknowledge the impact of the issue, which makes it difficult to harmonize DApps with legal contexts like GDPR.

Lack of trust in transactions has one cause in vulnerabilities of mechanisms interfering with the transaction flow against censorship and attacks. We already mentioned in Section 3 the issue of sequencer centralization in the rollups case and respectively of the private mempools used by ERC-4337 bundlers. In addition, most optimistic rollups do not have a working implementation of decentralized fraud proofs or this is limited to whitelisted actors. These are known issues, for which solutions are under development, but yet represent a factor that discourages implementation of DApps.

We summarize the range of the above DApp integration concerns in Table 1. Independent of these, a distinct aspect is the use of tokens, which is found in the majority of the DApps in the dominating areas of fintech and games. This seems to exert a driving force in technological advancements. An example is meta-transactions patterns, which were initially driven by the need of users lacking the native Ether currency to pay for transactions with tokens. This is reflected also in ERC-4337, which specifically mentions this use case. Also, many development frameworks see a main use of ERC-4337 in the context of smart contract wallets, which have a primary role in storing and managing tokens on behalf of users [9, 29].

While undoubtedly a necessity in many scenarios, we believe that this token dominance in the focus of the Ethereum community is an indirect cause of the limited DApps range. E-gov examples, like the land registry or voting applications, would fit with an implementation where a government entity would pay the fees

**Table 1: Summary of DApp integration concerns for rollups and ERC-4337 compared to typical Ethereum use**

| Technology | Transaction Flow Changes | Smart Contract Changes | Transaction Fraud Proofs | Added Centralization |
|---|---|---|---|---|
| Optimistic rollups | Yes | Not mandatory | Required | Sequencer traffic |
| Zk-rollups (w/o zkEVM) | Yes | Limited expressiveness | Not required | Sequencer traffic |
| Zk-rollups (with zkEVM) | Yes | Not mandatory | Not required | Sequencer traffic |
| ERC-4337 | Yes | Specific constructs | N/A | Private mempools |

on behalf of citizens who are mostly unaware of the blockchain technology, and respectively of tokens. In such contexts, the development of patterns like using the paymaster in ERC-4337 for transaction sponsorship is of much more interest than integration as a smart contract wallet, which seems to be a dominating trend. Another case is zk-rollups, which were mostly dedicated to token transactions in their initial form until the recent development of zkEVM based solutions. We argue that maintaining a more general perspective of technology than token focused will help towards an increase in the DApp range.

## 5 CONCLUSION

In this paper, we provided an overview of rollups and account abstraction, significant mechanisms that impact the Ethereum transaction flow to smart contracts, with the potential of increasing DApps range and adoption. We argued that these mechanisms increase the complexity of Ethereum DApps engineering, but their utility in lowering the transaction fees is essential. In our discussion we also referred to the trust issues these solutions still present. Considering the current trends in development, we think that a future direction could be AI driven analysis of smart contract functionality and selective choices in a DApp implementation. For instance, smart contract functions detected to have a high security sensitivity could be selectively sent as normal transactions, while less sensitive functions could be called as User Operations via a bundler, saving fees. A similar idea could be investigated also in relation to rollups. Transactions for which an AI analysis renders an acceptable tradeoff between security and costs could be grouped in smart contracts to be deployed on L2 rollups, while the rest would be kept on L1. An AI based solution used in the above scenario could most probably employ existing verification and analysis tools used for detecting problems in smart contract code. This approach is obviously a subject for a more complex discussion, but we see it as a direction worth exploring.

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
