# OpenReview forum: "Can Smart Contracts Become Smart? An Overview of Transaction Impact on Ethereum DApp Engineering"
_ACM.org/Middleware/Workshop/DICG — DICG 2023_

### Official Review · Reviewer_stFo · 2023-10-19
**Relevant review, but lacking motivation and systematization**

**Rating:** 6
**Confidence:** 3

**Review:**

The paper reviews recent technologies in the Ethereum ecosystem and their potential to increase adoption of DApps for applications in other domains, i.e. not just games and finance.

The paper especially reviews rollups, which allow to reduce fees by executing transactions off-chain, and the account abstraction, which allows third parties, e.g. application providers, to fund transactions.
Besides advantages for users, the authors also focus on the overhead these technologies impose on application developers.

The paper also presents a case study investigating the impact optimistic rollups have on transaction fees.

The paper reviews recent tools and frameworks which are relevant to the workshop. However, a more systematic comparison of the different technologies is missing. Also, I do not see sufficient argument that the presented technologies may increase adoption. For example, given prices for notarial land owner registration, I doubt that a transaction fee of 8$ is preventing the implementation of a land registry on ethereum.

Comments for the authors:

 Reviewing and systematizing recent solutions for rollups and account abstraction is a useful contribution. However, your paper does not show that these technologies may increase adoption.
 I believe a better way to present your work would be as a review of recent technologies.

Also, it would be nice if you could present the different technologies with their potential, drawbacks, and maturity, … in a table.

---

### Official Review · Reviewer_b5wQ · 2023-10-22
**Smart accounts are too smart for dapps**

**Rating:** 6
**Confidence:** 4

**Review:**

## Summary

In this work the authors describe the entanglement of transaction cost, L2s, and accessibility in Ethereum. It is argued that accessibility is currently being addressed by the development of scaling solutions, such as rollups, and through simplifications of the transaction flow, made possible by account abstraction. Ultimately the authors find many shortcomings of these systems and the ability for the entire stack to continuously adapt to changing implementations.

## Strengths

This work does an excellent job illuminating the state of ERC-4337 and its impact on on the flow of transaction data in Ethereum. It shows a strong understanding of the nuance which arises with account abstraction, specifically the difficulty integrating abstract and changing definitions of accounts into dapps.

The discussion of ERC-4337 and rollup sequencers together is an important and interesting point of analysis with respect to accessibility. The unification of many of these protocols is happening right now and was discussed by Vitalik Buterin in his post ["Endgame"](https://vitalik.ca/general/2021/12/06/endgame.html) as early as 2021. The authors of ERC-4337 are also [beginning to see](https://vitalik.ca/general/2023/09/30/enshrinement.html#enshrining-erc-4337) the limits of developing account abstraction at the application level and are looking to integrate it with the core protocol.

## Constructive Feedback

Unfortunately the strong section on ERC-4337 is overshadowed a bit by the lack of focus of the rest of the paper. The title of the paper goes mostly unaddressed in the work, which is seems more interested in investigating why smart contracts are not yet ubiquitous in society despite their many interesting use cases. It is noted in the first section in fact that it isn't that smart contracts are not smart enough, it's that they're too expensive.

The use case evaluation also fails to lead the reviewer towards any meaningful conclusion beyond L2 providing a cheaper environment for smart contract execution.

---

### Official Review · Reviewer_jiqU · 2023-10-28
**Interesting but not complete**

**Rating:** 5
**Confidence:** 5

**Review:**

This paper proposes a discussion on the challenges to design dApp (web application that use blockchain as a backend) for application that would touch a wider audience than current usage which is mainly restricted to decentralised finance (DeFi). The paper highlights two main problems: the cost of transaction (in native token of the backend blockchain) and the difficulty to acquire native tokens for non deFi users. In a second part of the paper the authors only address the first issue via a case study of a smart contract managing custody of land registries (so using blockchain as a notary service). The authors claim they have deployed the smart contract on Optimism (an L2 layer of ethereum that ensures low transaction fees by pushing on L1 transaction proofs rather than execution). Gas cost comparison then show that L2 layer offers a good solution to transaction fees.

The paper addresses an interesting topic namely the future of blockchain beyond speculative DeFi usage. The difficulty for the development of dApps is well known but it is worth presenting to an audience who might not be familiar with blockchains.

My main issue with the paper is that, although it claims to have deployed a smart contract on Optimism, it does not provide the deployment address nor the code of the smart contract. A public link to test dApp is also needed.

From a methodology point of view a paragraph explaining how the authors partitioned the transaction cost on Optimism between the L1/L2 gas cost would be a plus (methodology of Figure 1).